# Collagen Scaffolds Treated by Hydrogen Peroxide for Cell Cultivation

**DOI:** 10.3390/polym13234134

**Published:** 2021-11-26

**Authors:** Yuliya Nashchekina, Pavel Nikonov, Nataliya Mikhailova, Alexey Nashchekin

**Affiliations:** 1Center of Cell Technologies, Institute of Cytology of the Russian Academy of Sciences, Tikhoretsky Pr. 4, 194064 St. Petersburg, Russia; pashka2316@mail.ru (P.N.); natmik@mail.ru (N.M.); 2Laboratory “Materials and Structures of Solid State Electronics”, Ioffe Institute, Polytekhnicheskaya Str., 26, 194021 St. Petersburg, Russia

**Keywords:** collagen fibrils, hydrogen peroxide, scanning electron microscopy, adipogenic stromal cells, MG-63 osteosarcoma cell line, and A-431 epidermoid carcinoma

## Abstract

Collagen in the body is exposed to a range of influences, including free radicals, which can lead to a significant change in its structure. Modeling such an effect on collagen fibrils will allow one to get a native structure in vitro, which is important for modern tissue engineering. The aim of this work is to study the effect of free radicals on a solution of hydrogen peroxide with a concentration of 0.006–0.15% on the structure of collagen fibrils in vitro, and the response of cells to such treatment. SEM measurements show a decrease in the diameter of the collagen fibrils with an increase in the concentration of hydrogen peroxide. Such treatment also leads to an increase in the wetting angle of the collagen surface. Fourier transform infrared spectroscopy demonstrates a decrease in the signal with wave number 1084 cm^−1^ due to the detachment of glucose and galactose linked to hydroxylysine, connected to the collagen molecule through the -C-O-C- group. During the first day of cultivating ASCs, MG-63, and A-431 cells, an increase in cell adhesion on collagen fibrils treated with H_2_O_2_ (0.015, 0.03%) was observed. Thus the effect of H_2_O_2_ on biologically relevant extracellular matrices for the formation of collagen scaffolds was shown.

## 1. Introduction

The organization of collagen in the extracellular matrix serves as a determinant of tissue architecture [1]. There are 29 known collagen sub-types that contribute to tissue structure and function [2]. All collagen sub-types assemble into triple-helical structures formed from three left-handed α-helices chains that twist around each other into right-handed tropocollagen chains [3]. Such triple helical structures are named collagen molecular or monomer. Each α-helix consists of Gly-X-Y amino acid repeats that promote collagen fibril self-assembly and provide sites for inter-molecular and interfibrillar cross-linking [4]. Collagen monomers are molecules with ~300 nm length and ~1.36 nm diameter in solution, with an effective packing diameter of 1.52 nm [5]. Under certain conditions, collagen molecules assemble to fibrils with a wide variety of diameters. The narrowest collagen fibrils occur in vitreous humor, where they have a diameter ~10.5–12.0 nm [6], and cartilage, where collagen fibril has a diameter ~16–30 nm [7]. Thin but mechanically strong fibrils (~25 nm) can be found in the cornea, where they are organized in an exquisite orthogonal lattice [8]. Type I-collagen molecules are staggered by 67 nm against each other, resulting in the electron microscopically visible, classical D-banding of collagen fibrils. The fibrils supply attachment sites for a broad range of macromolecules such as fibronectin, proteoglycans, and cell surface receptors, including integrins and discoidin domain-containing receptors [9].

Much is known about the diameter of fibrils in tissues, but relatively little is known about fibril structure and length. Collagen fibrils are arranged in exquisite three-dimensional architectures in vivo, including orthogonal lattices in the cornea, parallel bundles in tendons and ligaments, basket weaves in the skin, and concentric weaves in bone and blood vessels. How the fibrils collect, how diameter and length are regulated, how other molecules attach to fibril surfaces, and how the multiscale organization is achieved are questions for which there are insufficient answers. The diameter, structure, and length of collagen fibrils in the different tissues change under various factors during the life of the body. Collagen molecules are also hydroxylated at specific prolyl hydroxylases or lysyl hydroxylases [10].

Collagen is a major component of connective tissue, a major meeting point of metabolic and catabolic reactions of tissues, and a large platform of signaling that regulates them [11]. One of the most general and significant processes is redox stress involved free radicals, which are atoms or molecules which contain one or more unpaired electron(s). The unpaired electron(s) give specific chemical properties to these molecules, such as the capacity to subtract electrons from other compounds to obtain stability [12]. Free radicals are either positively or negatively charged or electrically neutral molecules. The free radical theory postulates that the aging process is primarily a consequence of aerobic metabolism, which produces ROS in excess of cellular antioxidant defenses [13]. It is proposed that ROS oxidize cellular constituents such as proteins, nucleic acids, and lipids.

Redox reactions may be such oxidations as reductions. Oxidation is the gain of oxygen (O_2_), and the substance loses an electron, while reduction is the loss of O_2_, the gain of an electron, or the gain of hydrogen [14]. Reactive oxygen species are small molecules formed by partial reduction of molecular O_2_. ROS plays an important role in pathologic conditions such as cancer, heart diseases, diabetes, and of course, aging. Singlet oxygen (1O_2_), O_2_●− and hydrogen peroxide (H_2_O_2_) are the primary ROS products generated after the partial reduction of O_2_, while hydroxyl radicals (OH●) are generated in subsequent reactions. H_2_O_2_ is next to the superoxide anion and hydroxyl radical, a key member of the class of reactive oxygen species, which are by-products of cellular metabolism, including protein folding. It is an important metabolite involved in most of the redox metabolism reactions and processes of the cells and tissue [15,16], in various signal transduction pathways, and cell fate decisions. Depending on its intracellular concentration and localization, H_2_O_2_ exhibits either pro- or anti-apoptotic activities. In contrast to normal cells, cancer cells are characterized by an increased H_2_O_2_ production rate and an impaired redox balance, thus affecting the microenvironment and proteins of the extracellular matrix, collagen in particular. The effect of hydrogen peroxides on the structural properties of collagen in vitro as well as in vivo is still poorly understood, but studies on the effect of solution with high concentrations of hydrogen peroxide on collagen structure can be found in the literature. For example, solutions with 1–5% hydrogen peroxide are used in dental whitening treatments [17]. After exposure to 1% to 5% peroxide solutions, collagen fibers form agglomerates, and the rupture of intramolecular bonds occurs, leading to collagen secondary structure loss and degradation. These results show that H_2_O_2_ solutions have a significant effect on collagen structure. However, the concentration of hydrogen peroxide in the body is much lower, so it can be assumed that the destruction of collagen in vivo occurs to a much lesser extent. Moreover, the degree of destruction will depend on the state of the organism, age, etc.

Thus, collagen fibrils treated by solution with a low concentration of H_2_O_2_ will have a more native structure than intact collagen fibrils formed in vitro. Since collagen has a complex structural hierarchy with structural and biological features impacting at different length scales, it is important to study the interplay between collagen architecture (intact state: collagen fibrils forming in vitro vs. denatured state: after interaction with hydrogen peroxide) and oxidative stress under the influence of hydrogen peroxide. The goal of the work was to identify how hydrogen peroxide in solution with a low concentration influences the structure of collagen fibrils and how its structure will influence the functional activity of various tissue origin cells. The study of the interaction of cells with collagen fibrils before and after treatment with peroxide solutions with different concentrations is of both applied and fundamental interest.

## 2. Materials and Methods

### 2.1. Collagen Fibril Formation and Modification

Collagen was isolated from rat tail tendons by acid extraction. Collagen fibrils were formed by dissolving a protein in a 0.01% solution of acetic acid (Reactiv, Saint-Petersburg, Russia) to a concentration of 2 mg/mL. Subsequently, 1 M of KH_2_PO_4_ (Reactiv, Saint-Petersburg, Russia) was added to a final salt concentration of 20 mM and applied to a coverslip [18]. A solution of collagen and salt was kept for 15 min in an ammonia atmosphere. After that, samples were washed with Phosphate Buffered Saline (PBS) (Biolot, Saint-Petersburg, Russia).

To treat collagen fibrils with a solution of hydrogen peroxide (HP) with various concentrations of peroxide, coverslips with collagen were kept for 30 min in a solution of peroxide at room temperature, then the peroxide was removed, and the collagen fibrils were washed 3 times with a phosphate buffer solution (pH 7.4).

### 2.2. Scanning Electron Microscopy

The structure of the collagen fibrils before and after hydrogen peroxide modification was evaluated using a JSM-7001F (Jeol, Tokyo, Japan) scanning electron microscope (SEM).

### 2.3. Water Contact Angle

The static contact angles of water were measured at room temperature using the sessile drop method. The essence of the measurement was as follows. Collagen fibrils before and after hydrogen peroxide treatment were glued onto a glass slide. Then, 15 μL of distilled water was applied to the test sample with a special automatic dispenser. Using the device’s camera, the droplet shape on the test sample was evaluated, and the contact angle was calculated.

### 2.4. FTIR

The collagen fibrils before and after hydrogen peroxide treatment were analyzed using a Fourier transform infrared (FTIR) spectrometer IR Prestige-21 (Shimadzu, Tokyo, Japan), in transmission mode, in the 4000–600 cm^−1^ range, and with spectral resolution 2 cm^−1^.

### 2.5. Cell Cultivation

Adipogenic stromal cells (ASCs) were isolated from the abdominal visceral and subcutaneous adipose tissue of young (16 weeks) New Zealand white rabbits. Rabbit ASCs were isolated according to the protocol of Jung et al. [19]. ASCs were cultivated in α-minimum essential medium (α-MEM; Lonza, St. Louis, MO, USA) supplemented with 10% fetal bovine serum (FBS; HyClone, St. Louis, MO, USA), 100 U/mL penicillin (Sigma-Aldrich, Steinheim, Germany), and 100 mg/mL streptomycin (Sigma-Aldrich, Steinheim, Germany). Cells were cultivated in a CO_2_ incubator with an atmosphere of 5% CO_2_ content at 37 °C. For our experiments, cells were used following 2–6 passages, and cells were seeded in Petri dishes at a concentration of 1 × 10^6^ cells/cm^2^.

In vitro biocompatibility was also examined using the MG-63 osteosarcoma cell line and A-431 epidermoid carcinoma. The cell lines were obtained from the Vertebrate Cell Culture Collection (Institute of Cytology RAS, St-Petersburg, Russia). The cells were cultured in polystyrene flasks in DMEM (A-431) or EMEM (MG-63) supplemented with 10% FBS, 1% penicillin/streptomycin (Sigma-Aldrich, Darmstadt, Germany) at 37 °C in a humidified atmosphere of 5% CO_2_ in air. Sub-confluent cells were passaged by using trypsin– EDTA (0.25% (*w*/*v*) trypsin, 1 mM EDTA).

### 2.6. Fluorescence Staining of Cells

Cells were fluorescence stained by Rhodamine phalloidin in order to study the effects of collagen fibrils before and after hydrogen peroxide treatment on cell adhesion. Pure glass was used as a positive control, while unmodified collagen fibrils were the negative control. A precise description of the technique for fluorescent staining of cells was described in our previously published work [20]. Briefly, staining was performed as follows. After the cultivation period, the medium was removed, and the adherent cells were washed with PBS, fixed with a 4% formaldehyde solution (Sigma-Aldrich, Saint Louis, MO, USA). Next, a detergent solution was added to the cells. Rhodamine phalloidin (Thermo Fisher Scientific, Carlsbad, CA, USA) was used to stain the actin, and DAPI (ab104139; Abcam, Cambridge, MA, USA) was used to stain the nuclei. The cytoskeleton organization was analyzed using a confocal microscope Olympus FV3000 (Olympus Corporation, Tokyo, Japan).

### 2.7. Cell Counts

To study the effects of collagen fibrils before and after hydrogen peroxide treatment on cellular adhesion, a number of cells were cultured for 1 h or 1 day. Five different pictures of fields on each sample were used at a wavelength of 365 nm (DAPI) by a fluorescence microscope Pascal (Carl Zeiss Jena GmbH, Jena, Germany). The ImageJ program was used to count the nuclei in each picture.

### 2.8. Cell Viability

The MTT assay was employed to assess the cell viability and proliferation of the ASCs, A-549, and MG-63 on interaction with the collagen fibrils. This is a colorimetric assay measuring the reduction of yellow 3-(4,5-dimethythiazol-2-yl)-2,5-diphenyl tetrazolium bromide (MTT) (Sigma, St. Louis, MO, USA) substrate to an insoluble purple formazan product by mitochondrial succinate dehydrogenase enzyme. The MTT enters the cells, where it is reduced to an insoluble, dark purple-colored formazan product. As a control for dispersions, cells were cultured in a culture medium in an adhesive 96-well plate, while collagen fibrils before hydrogen peroxide treatment were considered as a control for collagen fibrils after hydrogen peroxide treatment. After 3 days of incubation, MTT solution was added to each well and incubated for 2 h at 37 °C with 5% CO_2_. After the incubation time, the above solution was discarded, and the colored formazan crystals formed were solubilized by adding 50 µL of dimethyl sulfoxide (Sigma, St. Louis, MO, USA). The absorbance was read at 570 nm in a multiwell plate reader (Thermo Fisher Multiscan Labsystems, Waltham, MA, USA).

### 2.9. Statistical Analysis

All experiments were performed in 3–5 replicates. The t-test was performed by using Microsoft Excel Software to analyze the statistically significant differences between specific samples. Samples were considered to be statistically important with a *p* < 0.05.

## 3. Results and Discussion

### 3.1. Scanning Electron Microscopy (SEM)

Collagen fibril scaffolds have always attracted the attention of researchers for cell culture and transplantation. A large number of studies and modern approaches are aimed at developing methods for the formation of fibrils of various diameters, orientations, and structures [21].

We formed fibrils on the basis of type I collagen extracted from the tendons of rat tails. The structure of the resulting fibrils was characterized using SEM. The structure of formed collagen fibrils was identical to the native form (Figure 1a).

The structure of the fibril has a certain periodicity of light and dark rings. This structure is due to the peculiarities of the molecular structure of the fibril. Such a model was first proposed in 1963 by Hodge and Petruska, who envisaged a bidimensional stack of five collagen molecules aligned parallel to one another with a staggering of about 67 nm [22]. This longitudinal or axial stagger represents the characteristic D-periodicity of the fibrils, which is the sum of the gap and overlap regions between collagen molecules. This model was later confirmed by more modern instrumental methods, such as transmission electron microscopy [23] and X-ray diffraction [24]. In SEM imaging, we can visualize the bright and dark gap regions. This color gradient is due to the peculiarities of the molecular structure of collagen fibrils. The molecule length (≈300 nm) is about 4.4–4.5D. A fibril contains an overlap or high-electron density regions (about 0.46D long) where side-by-side overlapping of adjacent triple helices occurs as well as a gap or low-electron density regions (about 0.54D long) with some space between the ends of the longitudinally lined-up molecules [3,25].

The SEM results demonstrated that treatment of native collagen fibrils with the hydrogen peroxide solution leads to a decrease in their diameter (Figure 1). Moreover, it is clearly seen that the fibrils become thinner in higher concentrations of hydrogen peroxide. It should be noted that before treatment with a hydrogen peroxide solution, all fibrils are of practically the same diameter, while after treatment with the solution, even a low concentration (0.006%) leads to the appearance of thinner fibrils along with fibrils whose diameter has not changed (Figure 1b). When collagen fibrils are treated with a peroxide solution with a maximum concentration (0.15%), the diameter of almost all fibrils decreases by a factor of 2 (Figure 1e). According to the literature, the diameter of collagen fibrils is affected by the presence of free and bound water that surrounds the triple helix [3,22]. It can be assumed that the decrease in the diameter of the fibrils is caused not by the removal of water around the fibril but by the oxidative effect of hydrogen peroxide [26].

The diameter of single triple helices depends on the degree of glycosylation, i.e., galactosylation and glucosyl-galactosylation of hydroxylysine residues [27]. A higher concentration of peroxide leads to a lower degree of glycosylation of hydroxylysine. Differences in the degree of glycosylation are a way to physiologically regulate fibril organization, as occurs in corneal type I collagen [28,29]. The degree of hydroxylation of proline and lysine is difficult to assess using SEM.

### 3.2. Water Contact Angle

The wettability (hydrophobicity and hydrophilicity) of cell adhesion surfaces can affect surface protein adsorption and cell adhesion. According to previous studies of other researchers, cells are more likely to adhere to hydrophilic surfaces. The wettability of a surface is greatly affected by the surface functional groups, the surface roughness of the material, and so on [30].

When studying the hydrophilicity of collagen fibrils before and after treatment with the hydrogen peroxide solution, a significant decrease to 40 degrees for all samples in the contact angle was found (Figure 2). This is 20 degrees less compared to untreated collagen fibrils. This result is also due to the oxidizing ability of hydrogen peroxide. An increase in the hydrophilicity of collagen fibrils may be due to the appearance of new hydroxyl groups formed as a result of treatment with a hydrogen peroxide solution.

### 3.3. Fourier Transform Infrared (FTIR) Spectroscopy

FTIR spectra of collagen fibril exhibit distinct signals at 3330–3325 cm^–1^ (amide A), 3080 cm^–1^ (amide B), 1650 cm^–1^ (amide I), and 1550 cm^–1^ (amide II). Figure 3 shows the spectrum of collagen fibrils before and after treatment with peroxide solutions with different concentrations. According to the results obtained, the signal intensity of untreated collagen fibrils is higher in almost the entire spectral region than in the treated collagen fibrils (Figure 3).

The amide A band of collagen is associated with NH-stretching [31]. The frequency of the amide A bands depends on the conformation of collagen; the less structural order the protein has, the lower the frequency of these amide bands [32]. After exposure to hydrogen peroxide, a band shift of 3324 cm^−1^ was observed. The shift of the band can be caused by cleavage of the amide bond of the main or side chain [33]. Similar destruction of collagen fibrils was observed upon exposure to UV irradiation [34]. It has been demonstrated that UV irradiation is capable of breaking C-N bonds and causing peptide bonds to break. According to the results of other authors, the shift of amide A towards a lower spectral frequency is due to the destruction of hydrogen bonds, which leads to a change in the structural order of collagen [33].

The amide B (asymmetrical stretch of CH_2_) bands were found at 2925 cm^−1^ [35]. Collagen displays bands at 1647, 1547, and 1240 cm^−1^, which are characteristic of the amide I, II, and III bands of collagen. The amide I absorption arises predominantly from protein amide C=O stretching vibrations [36]. The amide II absorption is made up of amide N–H bending vibrations and C–N stretching vibrations. The amide III peak is complex, consisting of components from C–N stretching and N–H in-plane bending from amide linkages, as well as absorptions arising from wagging vibrations from CH_2_ groups from the glycine backbone and proline side-chains.

Particular attention should be paid to the signal at 1084 cm^−1^, the intensity of which significantly decreases after the treatment of collagen fibrils with hydrogen peroxide. The region of the spectrum from 1005–1100 cm^−1^ is characteristic for ʋ(C-O-C) [37]. We assume that the decrease in the signal with wave number 1084 cm^−1^ is due to the detachment of glucose and galactose linked to hydroxylysine, which are connected to the collagen molecule through the -C-O-C- group. As a result, a hydroxyl group is formed on hydroxylysine (Figure 4). Hydroxyl groups increase the hydrophilicity of collagen fibrils (Figure 2). Removal of glucose and galactose residuals also reduced the distance between collagen molecules and led to the collagen fibril diameter decrease that was demonstrated by SEM (Figure 1). We did not observe noticeable changes in the FTIR spectra (Figure 3), which allowed us to make the assumption that covalent bonds (aldol and aldimine cross-links) between collagen molecules are not destroyed.

### 3.4. In Vitro Biological Evaluation

Type I collagen is found in tissues such as skin, bones, cartilage, cornea, and blood vessels. When culturing cells in vitro, type I collagen is used as part of films, scaffolds, and for modifying culture vessels [38,39,40]. Different connective tissues are characterized not only by differences in the collagen fibril diameter but by tissue-specific suprafibrillar architectures, as seen by polarized light microscopy [41]. For example, in bone tissue, collagen fibrils are stiffened and hardened by infiltrating inorganic particles. The fraction of minerals laid out inside the fibrils appear to be tissue-dependent but is usually high (in most cases near 67%), and it reinforces the tissue by simultaneously increasing the tensile strength of fibrils, their elastic modulus, and the inter-fibrillar mechanical coupling [42].

The study of the interaction of cells with collagen fibrils before and after treatment with peroxide solutions with different concentrations is of both applied and fundamental interest. As mentioned earlier, in the process of the vital activity of the body, collagen is subjected to oxidative stress under the influence of free radicals. As a result of this effect, various biochemical processes occur, including the rupture of covalent intermolecular bonds and the glycolysis of oxylysine. The presence of active radicals can cause both denaturation and cross-linking of collagen. To study the effect of treating collagen fibrils with peroxide solutions of various concentrations, we used cells of various tissue origin, including primary culture (ASCs) and transformed cell lines. 

As a result of the experiment, it was demonstrated that within 1 h after sowing on a surface modified with collagen fibrils, more cells were attached compared to the positive control (glass) (Figure 5(Ia)). Treatment with a low concentration of a peroxide solution increased the adhesion of all types of studied cell cultures. The greatest increase in adhesion was observed in A-431 cells (Figure 5(Ii)). With an increase in the concentration of peroxide from 0.015% to 0.15%, these cells retained a high adhesion ability to collagen fibrils. However, an increase in the concentration of hydrogen peroxide leads to a decrease in the number of adhered MG-63 cells (Figure 5(Io–r)). High adhesion capacity, which is proportional to the concentration of hydrogen peroxide, is retained only in A-431 cells. It should be noted that the treatment of collagen fibrils with a peroxide solution with a low concentration affects not only the number of adhered cells but also the degree of flattening. Figure 5 shows that ASCs on collagen fibrils treated with a peroxide solution with a concentration of 0.006% have a well-organized cytoskeleton after 1 h of cultivation (Figure 5(Ic)). In Figure 5, individual stress actin fibrils can be observed. With an increase in peroxide concentration, the degree of spreading of ASCs decreases, and stress fibrils are concentrated near the nucleus (Figure 5(Ie,f)). A similar relationship was observed when fibroblasts were cultured on partially denatured collagen [43]. Fisher et al. demonstrated that human skin fibroblasts cultured on denatured collagen fibrils treated with matrix metalloproteinases are less spread and elongated compared to fibroblasts cultured on intact collagen fibrils [44]. A similar morphology of fibroblasts with a reduced cytoplasmic area was observed in fibroblasts in aged human skin in vivo [43]. A statistically significant increase in the number of adherent cells on collagen fibrils treated with hydrogen peroxide after 1 h of cultivation was observed only in A-431 cells (Figure 5II).

Within 1 day after sowing, the organization of the actin cytoskeleton of cells changes depending on the hydrogen peroxide concentration. It should be noted that ASC cells on collagen fibrils treated with a low concentration hydrogen peroxide solution have a more rounded shape compared to cells on unmodified collagen fibrils (Figure 6(Ic–f)). Cells on native collagen fibrils have a more elongated shape (Figure 6(Ib)). The number of adhered ASCs, A-431, and MG-63 cells after 1 day of cultivation is also higher on collagen fibrils treated with a peroxide solution compared to cells cultivated on unmodified collagen fibrils (Figure 6II). However, with an increase in the concentration of peroxide to 0.15%, the number of adhered cells decreases significantly. A similar picture was observed in ASCs and MG-63 cells.

In the study of cell proliferation on the studied surfaces, an increase in the proliferative activity of cells was observed in proportion to the increase in the concentration of peroxide solution only during the cultivation of A-431 cells (Figure 7). Sufficiently high cell proliferation was observed for ASCs. Treatment of collagen fibrils with a peroxide solution with a low concentration did not affect MG-63 cells proliferation. However, at a maximum peroxide concentration of 0.15%, cell proliferation decreased in all cases.

Collagen fragmentation, a reduction in total collagen, and decreased cell-collagen fiber interactions also characterize chronologically aged skin [44,45]. In our study, we observed an increase in the adhesion and proliferation of cells of various tissue origins on partially denatured collagen. Obviously, this is due to the fact that during the entire life of the body, including at a young age, the extracellular matrix is exposed to free radicals. As a result of this effect, collagen is partially denatured. It is the partial denaturation of intermolecular but not intramolecular bonds in the collagen fibril that provides more native properties of collagen fibrils.

## 4. Conclusions

Collagen is one of the main proteins of the extracellular matrix, which is widely used in the formation of scaffolds for cell culture and transplantation. The native structure of collagen fibrils has been studied by many physicochemical methods. Usually, when forming collagen scaffolds, researchers are guided by the native structure of collagen. However, in the body, collagen undergoes various chemical and structural changes under the influence of external factors, including free radicals. To obtain collagen fibrils whose properties would be maximally similar to collagen fibrils in the body, we treated them with a solution of hydrogen peroxide. It has been shown that such treatment actually leads to structural and functional changes in the collagen fibril that increase the adhesion of ASCs and A-431 cells. It was also shown that treatment of collagen fibrils with a hydrogen peroxide solution increases the proliferative activity of ASCs and A-431 cells. We have demonstrated by FTIR spectroscopy that the treatment does not lead to the breaking of covalent bonds (aldol- and aldimine cross-links) between collagen molecules. A decrease in collagen fibrils’ diameter, as well as a decrease in the wetting angle and in the signal in the FTIR spectrum, characteristic of ʋ (C-O-C), allowed us to make the assumption that glucose and galactose are detached from the collagen molecule as a result of hydrogen peroxide solution treatment. In the future, we plan to carry out accurate measurements of the diameter of collagen fibrils after treatment with a peroxide solution and calculate the quantitative dependence of the destructive capacity of hydrogen peroxide on the diameter of collagen fibrils.

Thus, we assume that the treatment of collagen fibrils with a peroxide solution will simulate native collagen and form scaffolds whose properties would mimic the native extracellular matrix, which is subjected to oxidative stress in the body.

## Figures and Tables

**Figure 1 polymers-13-04134-f001:**
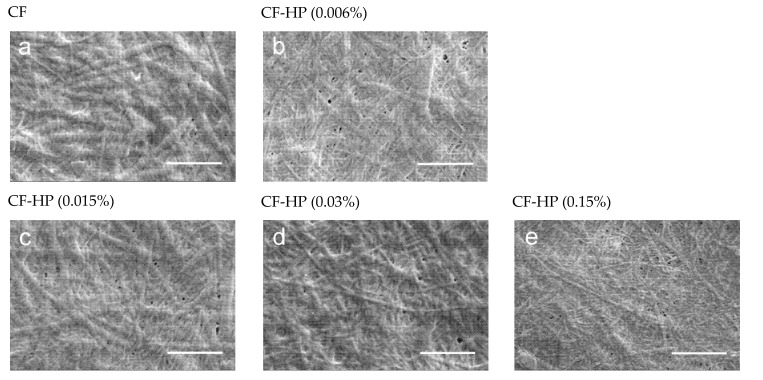
SEM images of collagen fibrils after different HP treatment: (**a**)—collagen fibrils before treatment; (**b**)—collagen fibrils after HP treatment with concentration 0.006%; (**c**)—fibrils after HP treatment with concentration 0.015%; (**d**)—fibrils after HP treatment with concentration 0.03%; (**e**)—fibrils after HP treatment with concentration 0.15%. Scale bar 1 µm.

**Figure 2 polymers-13-04134-f002:**
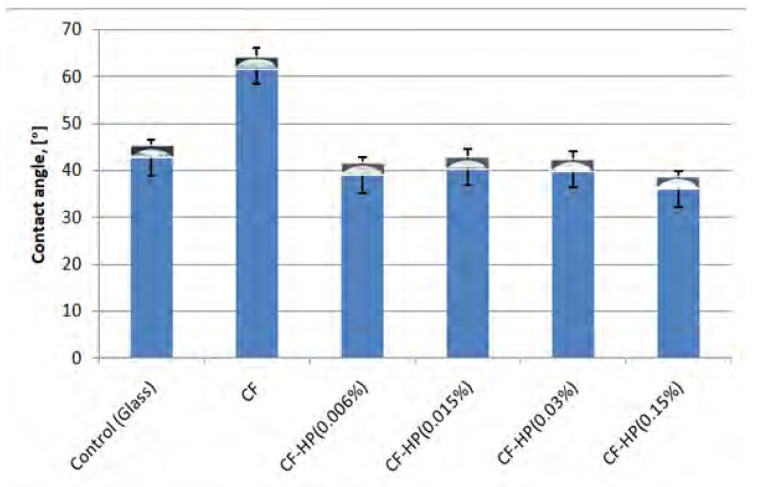
Contact angles of the collagen fibrils after different HP treatment: CF—collagen fibrils before treatment; CF-HP (0.006%)—collagen fibrils after HP treatment with concentration 0.006%; CF-HP (0.015%)—collagen fibrils after HP treatment with concentration 0.015%; CF-HP (0.03%)—collagen fibrils after HP treatment with concentration 0.03%; CF-HP (0.15%)—collagen fibrils after HP treatment with concentration 0.15%.

**Figure 3 polymers-13-04134-f003:**
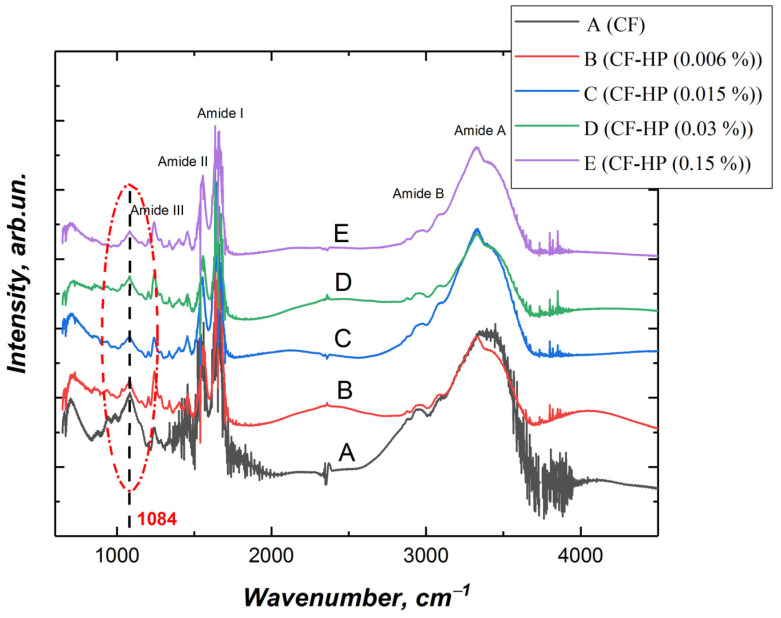
FTIR spectra of collagen fibrils after different HP treatment: CF—collagen fibrils before treatment; CF-HP (0.006%)—collagen fibrils after HP treatment with concentration 0.006%; CF-HP (0.015%)—collagen fibrils after HP treatment with concentration 0.015%; CF-HP (0.03%)—collagen fibrils after HP treatment with concentration 0.03%; CF-HP (0.15%)—collagen fibrils after HP treatment with concentration 0.15%.

**Figure 4 polymers-13-04134-f004:**
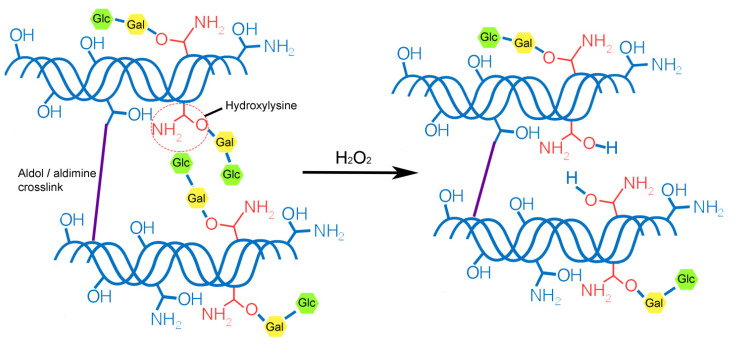
Schematic diagram of collagen fibril structure modification after hydroxide peroxide treatment. (Gal—galactose, Glc—glucose).

**Figure 5 polymers-13-04134-f005:**
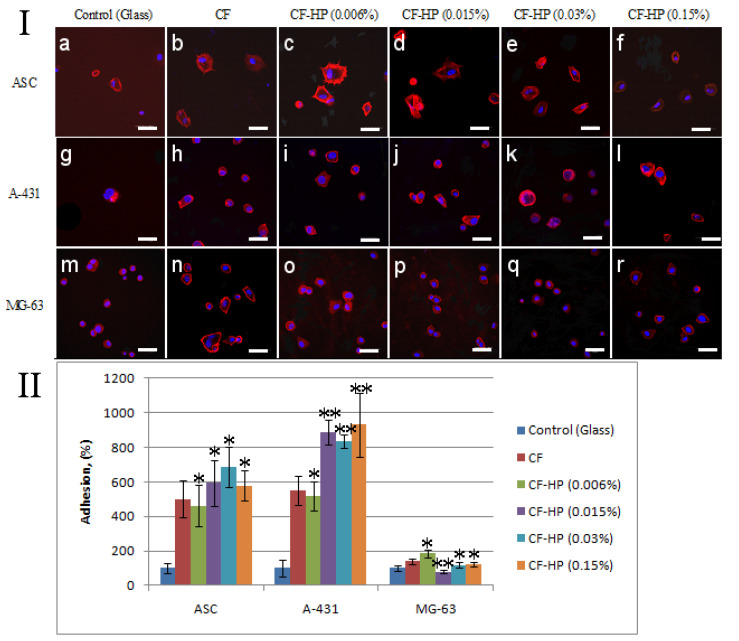
(**I**) Fluorescence micrographs of cells with actins (red) and nuclei (blue) stained after 1 h cultivation: (**a**) ASCs on the pure glass, (**b**) ASCs on the pristine CF, (**c**) ASCs on CF after HP treatment with concentration 0.006%, (**d**) ASCs on CF after HP treatment with concentration 0.015%, (**e**) ASCs on CF after HP treatment with concentration 0.03%, (**f**) ASCs on CF after HP treatment with concentration 0.15%, (**g**) A-431 on the pure glass, (**h**) A-431 on the pristine CF, (**i**) A-431 on CF after HP treatment with concentration 0.006%, (**j**) A-431 on CF after HP treatment with concentration 0.015%, (**k**) A-431 on CF after HP treatment with concentration 0.03%, (**l**) A-431 on CF after HP treatment with concentration 0.15%, (**m**) MG-63 on the pure glass, (**n**) MG-63 on the pristine CF, (**o**) MG-63 on CF after HP treatment with concentration 0.006%, (**p**) MG-63 on CF after HP treatment with concentration 0.015%, (**q**) MG-63 on CF after HP treatment with concentration 0.03%, (**r**) MG-63 on CF after HP treatment with concentration 0.15%. (**II**) Statistics of cell adhesion on surfaces with different functional groups after 1 h cultivation. (n = 5: *—*p* < 0.01, **—*p* < 0.05 compared with the unmodified CF.) Staining—rhodamine-phalloidin (red), 40,6-diamidino-2-phenylindole (DAPI) (blue). Scale bar 50 µm.

**Figure 6 polymers-13-04134-f006:**
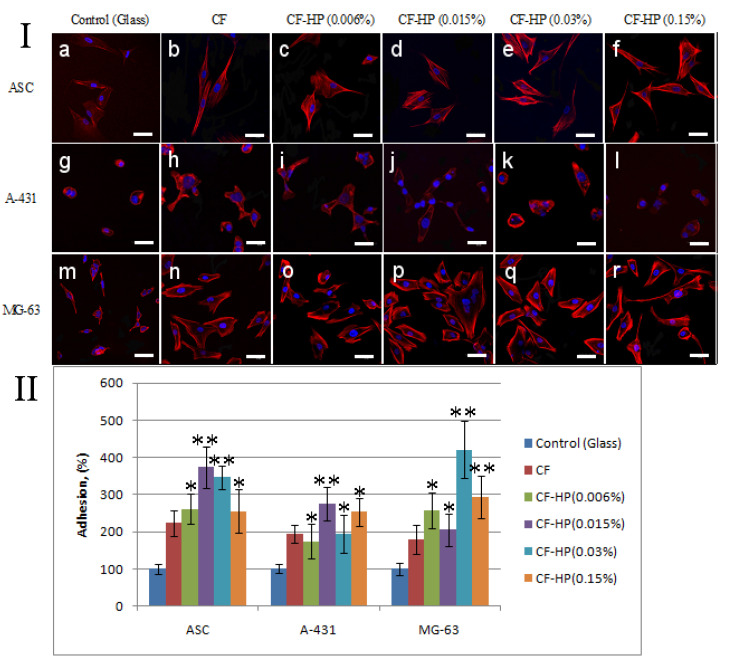
(**I**) Fluorescence micrographs of cells with actins (red) and nuclei (blue) stained after 1 day cultivation: (**a**) ASCs on the pure glass, (**b**) ASCs on the pristine CF, (**c**) ASCs on CF after HP treatment with concentration 0.006%, (**d**) ASCs on CF after HP treatment with concentration 0.015%, (**e**) ASCs on CF after HP treatment with concentration 0.03%, (**f**) ASCs on CF after HP treatment with concentration 0.15%, (**g**) A-431 on the pure glass, (**h**) A-431 on the pristine CF, (**i**) A-431 on CF after HP treatment with concentration 0.006%, (**j**) A-431 on CF after HP treatment with concentration 0.015%, (**k**) A-431 on CF after HP treatment with concentration 0.03%, (**l**) A-431 on CF after HP treatment with concentration 0.15%, (**m**) MG-63 on the pure glass, (**n**) MG-63 on the pristine CF, (**o**) MG-63 on CF after HP treatment with concentration 0.006%, (**p**) MG-63 on CF after HP treatment with concentration 0.015%, (**q**) MG-63 on CF after HP treatment with concentration 0.03%, (**r**) MG-63 on CF after HP treatment with concentration 0.15%. (**II**) Statistics of cell adhesion on surfaces with different functional groups after 1 h cultivation. (n = 5: *—*p* < 0.01, **—*p* < 0.05 compared with the unmodified CF.) Staining—rhodamine-phalloidin (red), 40,6-diamidino-2-phenylindole (DAPI) (blue). Scale bar 50 µm.

**Figure 7 polymers-13-04134-f007:**
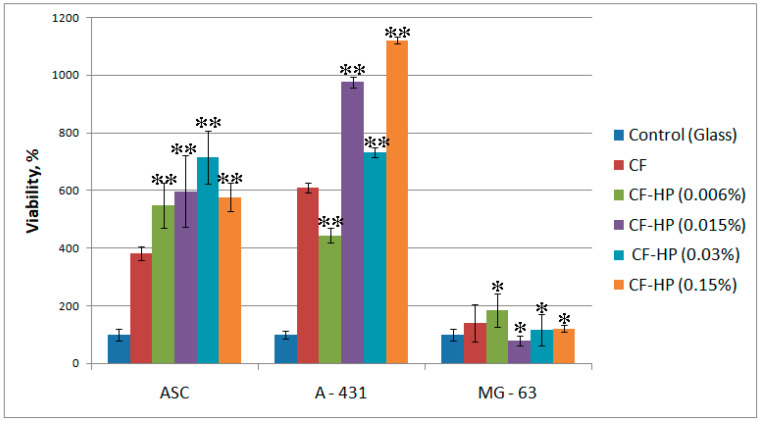
Citotoxity study (MTT-test) with ASCs, A-431 and MG-63 cells on collagen fibrils after different HP treatment: CF—collagen fibrils before treatment; CF-HP (0.006%)—collagen fibrils after HP treatment with concentration 0.006%; CF-HP (0.015%)—collagen fibrils after HP treatment with concentration 0.015%; CF-HP (0.03%)—collagen fibrils after HP treatment with concentration 0.03%; CF-HP (0.15%)—collagen fibrils after HP treatment with concentration 0.15%. (n = 5: *—*p* < 0.01, **—*p* < 0.05 compared with the unmodified CF).

## Data Availability

Data available upon request.

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
