# Peer review of "Collagen Scaffolds Treated by Hydrogen Peroxide for Cell Cultivation"

_polymers, 2021, doi:10.3390/polym13234134_

Round 1

Reviewer 1 Report

The paper “Collagen Scaffolds Treated by Hydrogen Peroxide for Cell Cultivation” Yuliya Nashchekina et al. reports the results of research on the influence of H2O2 on collagen fibers and the formation of collagen scaffolds.

The article deals with a very important issue, which is the influence of free radicals and their role in the formation and development of pathological changes and the aging of organisms. The authors narrowed their research to collagen fibers and three groups of cells (ASCs, MG-63, and A-431), and they focus on H2O2 concentrations closer to that found naturally in organisms. This approach definitely increases the value of the paper. The purpose of the paper is to answer the questions 1) how H2O2 in solution with a low concentration influence the structure of collagen fibrils and 2) how colagen structure will influence the functional activity of various tissue origin cells. Searching for answers, the authors analyze the data from several experimental techniques, SEM and fluorescence microscope, FTIR, they measure also the water contact angle and the cell viability. The authors shows that treatment of native collagen fibrils with the H2O2 solution leads to a decrease in their diameter – the higher concentrations the thinner collagen fibrils. The source of these structural changes is not the breaking of covalent bonds (aldol- and aldimine crosslinks) between collagen molecules, but detached of glucose and galactose. They shown that treatment of collagen fibrils with a hydrogen peroxide solution increases the proliferative activity of ASCs and A-431 cells.

Authors may consider improving the article by clarifying and expanding on the following points:

  • in chapter 2.3.it is worth specifying the size of the water droplets and the model used to determine the contact angle;
  • the sentences in lines 192-194 seem to duplicate the information from the introduction;
  • on the basis of the SEM photos (fig. 1), it is possible to measure the diameters of collagen fibers.The authors indicate that they plan to carry out a detailed study of this parameter in the future, but it seems to be worthwhile to provide at least the averaged results in this paper.The comment on the planned further steps of the work (lines 223-227) could be moved to Conclusions to indicate the way forward for the research;
  • in chapter 3.2 it is worth adding a sentence explaining and justifying this type of measurement: what is the purpose of such a measurement?what will it show?
  • an excerpt from chapter 3.4 (lines 309-317) is more suited to the introduction,
  • Please note that most of the references in the text mislead the reader to Figure 0 (e.g. l. 255, 291,292,334,337)- this must be corrected.

In my opinion article is prepared fairly, the presented results are new and according to the reviewer's knowledge they were not published before. The article present the results of a number of techniques and in my opinion the article should be published.

Author Response

Comments and Suggestions for Authors

The paper “Collagen Scaffolds Treated by Hydrogen Peroxide for Cell Cultivation” Yuliya Nashchekina et al. reports the results of research on the influence of H2O2 on collagen fibers and the formation of collagen scaffolds.

The article deals with a very important issue, which is the influence of free radicals and their role in the formation and development of pathological changes and the aging of organisms. The authors narrowed their research to collagen fibers and three groups of cells (ASCs, MG-63, and A-431), and they focus on H2O2 concentrations closer to that found naturally in organisms. This approach definitely increases the value of the paper. The purpose of the paper is to answer the questions 1) how H2Oin solution with a low concentration influence the structure of collagen fibrils and 2) how colagen structure will influence the functional activity of various tissue origin cells. Searching for answers, the authors analyze the data from several experimental techniques, SEM and fluorescence microscope, FTIR, they measure also the water contact angle and the cell viability. The authors shows that treatment of native collagen fibrils with the H2Osolution leads to a decrease in their diameter – the higher concentrations the thinner collagen fibrils. The source of these structural changes is not the breaking of covalent bonds (aldol- and aldimine crosslinks) between collagen molecules, but detached of glucose and galactose. They shown that treatment of collagen fibrils with a hydrogen peroxide solution increases the proliferative activity of ASCs and A-431 cells.

Response

Dear Reviewer 1, we are very grateful to you for manuscript review and valuable comments. We tried to take into account all your comments and recommendations and noticeably improve the manuscript. Thanks a lot.

Authors may consider improving the article by clarifying and expanding on the following points:

  • in chapter 2.3.it is worth specifying the size of the water droplets and the model used to determine the contact angle;

The static contact angles of water were measured using the sessile drop method. 15 μl of distilled water was applied to the test sample. On average, the droplet diameter was 7-9 mm, the height was 2-2.5 mm.

Most methods for wetting characterization can be classified into two main groups. In optical methods, the shape of a droplet is measured, whereas most other methods assess the force exerted by water on the solid. Owing to its versatility and ease of use, the optical method called sessile-drop goniometry is probably the most widely used (Huhtamäki, T., Tian, X., Korhonen, J.T. et al. Surface-wetting characterization using contact-angle measurements. Nat Protoc 13, 1521–1538 (2018). https://doi.org/10.1038/s41596-018-0003-z). Owing to its versatility and ease of use, the optical method called sessile-drop goniometry is probably the most widely used. 

  • the sentences in lines 192-194 seem to duplicate the information from the introduction;

We agree with this comment and exclude this information from the section.

  • on the basis of the SEM photos (fig. 1), it is possible to measure the diameters of collagen fibers. The authors indicate that they plan to carry out a detailed study of this parameter in the future, but it seems to be worthwhile to provide at least the averaged results in this paper. The comment on the planned further steps of the work (lines 223-227) could be moved to Conclusions to indicate the way forward for the research.

Measurements of the average thickness of fibrils on each of the samples were carried out in the free ImageJ program using the following method. In each image, three areas were selected in which the fibril structures are most clearly visible. A linear profile perpendicular to the fibrils was built on each of them. Five measurements of the fibril diameters were made on each profile, while the largest and smallest fibril diameters were not taken into account. All the obtained measurement results for each sample were averaged, and the mean square deviation was also calculated.

To obtain statistically reliable results, additional complex analysis of 2D images is required, which we plan to do in the next article.

  • in chapter 3.2 it is worth adding a sentence explaining and justifying this type of measurement: what is the purpose of such a measurement?what will it show?

«The wettability (hydrophobicity and hydrophilicity) of cell adhesion surfaces can affect surface protein adsorption and cell adhesion. According to previous studies of other researchers, cells are more likely to adhere to hydrophilic surfaces. The wettability of a surface is greatly affected by the surface functional groups, the surface roughness of the material, and so on. (Ayala R, Zhang C, Yang D, Hwang Y, Aung A, Shroff SS, et al. Engineering the cell-material interface for controlling stem cell adhesion, migration, and differentiation. Biomaterials. 2011;32:3700–11.)»

These sentences have been inserted in the section "3.2 Water contact angle"

  • an excerpt from chapter 3.4 (lines 309-317) is more suited to the introduction,

The authors, with the comments of the reviewer, have moved this section to the Introduction. We think that information about Young's modulus is unnecessary in this research. Sentence “The study of the interaction of cells with collagen fibrils before and after treatment with peroxide solutions with different concentrations is of both applied and fundamental interest.” we have moved to the section "Introduction"

  • Please note that most of the references in the text mislead the reader to Figure 0 (e.g. l. 255, 291,292,334,337)- this must be corrected.

The authors agree with the comments of the reviewer and made corrections to the text.

In my opinion article is prepared fairly, the presented results are new and according to the reviewer's knowledge they were not published before. The article present the results of a number of techniques and in my opinion the article should be published.

Reviewer 2 Report

Dear all,

Greetings

Please find enclosed my comments regarding paper

Referenced as: polymers-1469171

Titled: Collagen scaffolds treated by hydrogen peroxide for cell cultivation

 The authors have performed excellent paper for biological purpose, about collagen and the effect of free radical from H2O2, but this article can be accepted for publication in Polymers, after fixing all these recommendations (Minor Revisions)

1) Title: Ok

2) Abstract: please add the best rate of H2O2 on the collagen structure 3) Keywords: very long (IR)

4) Comments:

  • Figure 1 you used 0.03 then 0.15 in term of concentration why?
  • Figure 4 how did prove that this structure it's correct
  • Figure 5 you put so much information’s, is better to splited in two figures same remarques for Figure 6.

please update them add some of 2021 and 2022 and fellow template of the journal for all references

With regards

Author Response

(The authors gave the same response as above.)
